# Cellcano: supervised cell type identification for single cell ATAC-seq data

Wenjing Ma [1], Jiaying Lu [1] & Hao Wu [2,3] ✉

Computational cell type identification is a fundamental step in single-cell omics data analysis. Supervised celltyping methods have gained increasing popularity in single-cell RNA-seq data because of the superior performance and the availability of high-quality reference datasets. Recent technological advances in profiling chromatin accessibility at single-cell resolution (scATAC-seq) have brought new insights to the understanding of epigenetic heterogeneity. With continuous accumulation of scATAC-seq datasets, supervised celltyping method specifically designed for scATAC-seq is in urgent need. Here we develop Cellcano, a computational method based on a two-round supervised learning algorithm to identify cell types from scATAC-seq data. The method alleviates the distributional shift between reference and target data and improves the prediction performance. After systematically benchmarking Cellcano on 50 well-designed celltyping tasks from various datasets, we show that Cellcano is accurate, robust, and computationally efficient. Cellcano is well-documented and freely available at https://marvinquiet.github.io/Cellcano/.

The developments of single cell sequencing technologies have greatly enhanced the understanding of biological mechanisms in complex tissues. Among all single cell assays, single-cell RNA-sequencing (scRNA-seq) has been the most popular with over 1200 analytical tools developed[1]. In scRNA-seq data analysis, computational cell type identification based on gene expression values of individual cells (referred to as "celltyping" hereafter) is one of the most fundamental questions. Many celltyping methods are currently available[2–9] and several benchmark papers have been published[10–12]. These methods can be roughly categorized as supervised and unsupervised. According to benchmark studies, supervised celltyping methods have advantages over unsupervised ones in accuracy, robustness and scalability[13,14].

Gene expression can be regulated by several factors. Among them, chromatin accessibility is essential for the interaction between DNA and regulatory elements, and provides important information for understanding the transcriptional regulatory mechanism[15]. Recent years have also witnessed the shift from measuring chromatin accessibility in bulk samples to single-cell level by single-cell sequencing assay for transposase-accessible chromatin (scATAC-seq)[16]. Like in scRNA-seq, celltyping is also an important question in scATAC-seq data analysis. However, scATAC-seq data have certain characteristics that make the celltyping more difficult. First of all, scATAC-seq data are much sparser due to low read counts[17], which results in weaker signals for distinguishing cell types. Secondly, unlike scRNA-seq, feature space is not well-defined in scATAC-seq data, which poses difficulties in extracting useful information. The raw scATAC-seq data can be summarized to counts on genome-wide fixed-size bins, peaks representing the accessible regions, or genes[18]. Thus, the determination of feature space is an additional important step in scATAC-seq celltyping. Although it is possible to do celltyping through experimental procedures such as Fluorescence-activated cell sorting (FACS)[19] or leveraging information from multi-omics sequencing techniques such as SNARE-seq[20], these datasets are expensive and limited. Therefore,

[1]Department of Computer Science, Emory University, 400 Dowman Drive, Atlanta, GA 30322, USA. [2]Faculty of Computer Science and Control Engineering, Shenzhen Institute of Advanced Technology, Chinese Academy of Sciences, 1068 Xueyuan Avenue, Shenzhen University Town, Shenzhen 518055, P. R. China. [3]Department of Biostatistics and Bioinformatics, Rollins School of Public Health, Emory University, 1518 Clifton Road NE, Atlanta, GA 30322, USA. ✉e-mail: wuhao@siat.ac.cn

method specifically developed for scATAC-seq celltyping is in urgent need.

Most existing computational scATAC-seq celltyping methods are unsupervised and based on prior knowledge[21–24]. As of now, many methods have been developed for single-cell omics data integration while very limited methods have been specifically developed for scATAC-seq celltyping. Seurat[24] and scJoint[25] use scRNA-seq as reference to transfer cell labels to scATAC-seq. Due to the strong data distributional shift between different measurements, the two methods can significantly underperform. Signac[26], a recently developed end-to-end scATAC-seq data analysis pipeline, provides functions for scATAC-seq data integration and label transfer. EpiAnno was also published very recently to perform supervised celltyping in scATAC-seq using scATAC-seq as reference[27]. A major problem of Signac and EpiAnno is that they use read counts from called peaks as input, where the peaks are highly data dependent. Due to technical and biological artifacts, concordance of peaks can be low between reference and target[28], which would result in a loss of information and undesirable celltyping results. Additionally, EpiAnno is not computationally scalable for large datasets.

In this work, we develop a computational celltyping method for scATAC-seq, named Cellcano. Cellcano implements a two-round supervised learning algorithm. It first trains a multi-layer perceptron (MLP) on the reference dataset and predicts cell types in target data. From the prediction results, Cellcano selects some target cells that are considered well-predicted (referred to as anchors) to form a new training set. Next, Cellcano trains a self-Knowledge Distiller model (KD model)[29] on anchors using the predicted pseudo labels, and then apply the trained KD model to predict cell types for remaining non-anchor cells. Through extensive real data analyses, we demonstrate that Cellcano is significantly more accurate, computationally efficient, and scalable compared to existing methods. Cellcano is well-documented and freely available at https://marvinquiet.github.io/Cellcano/.

## Results

### The Cellcano framework

Cellcano uses gene-level summaries from the raw scATAC-seq data as inputs. Given the raw data, Cellcano incorporates ArchR[30] pipeline to process the raw data and obtain gene scores for both reference and target datasets (details in Methods section). The choice of the input is carefully investigated, and the results show that using gene scores provides good prediction accuracy and computational efficiency (details in later section). Then Cellcano applies F-test on reference gene scores to select cell-type-specific genes as features for model construction[31]. After obtaining the reference and target gene scores for the selected features, Cellcano adopts a two-round supervised cell-typing strategy, shown in Fig. 1. In the first round, Cellcano trains an MLP model with reference gene scores and predicts cell types in target data. If the target size is too small, Cellcano stops and returns the prediction results. When the target size is large enough (e.g., over 1000 cells), Cellcano performs another round of model training to improve the prediction results. The second round starts with selecting anchor cells. For that, we first calculate entropy for each cell based on the prediction probabilities from the first-round prediction and then select cells with lower entropies as anchors. The assumption is that the cells with lower prediction entropies are more likely to be accurately predicted. We carefully investigate the anchor cell properties and their impact on the prediction results (details in later section) and demonstrate that the assumption holds well in real data. We then use the anchors with their predicted cell types as new reference data to train another classifier to predict the non-anchor cells. Here, we use a KD model as the classifier since it works better when reference data have imperfect labels. The assumption in the second round is that the classifier trained on anchors (which are from the target data) can better capture the data distribution in the target dataset compared to the

classifier trained on the reference dataset, thus improve the prediction performance.

### Celltyping tasks

We collect and process four human peripheral blood mononuclear cells (PBMCs) datasets and two mouse brain datasets (Supplementary Table 1; details in Methods section). Among four human PBMCs datasets, one is cell-sorted by FACS and can be considered as gold standard. The cell types in other three datasets are annotated based on computational methods and prior biological knowledge, which are silver standard[32]. For the six datasets, we design 50 celltyping tasks (details in Supplementary Note 1; tasks listed in Supplementary Dataset 1 and 2), which comprehensively cover different real application scenarios. All results in following subsections are based on these tasks.

### The choice of using gene score as input

As mentioned before, scATAC-seq data can be represented in three different feature spaces: genome-wide fixed-size bins, peaks, and genes. Genome-wide fixed-size bins have a very large feature space, which poses heavy computational burden. The peaks are not pre-defined and require additional steps in calling and unifying peaks. More importantly, since the peaks will be different for each dataset, one cannot reuse a pre-trained prediction model for new target data. In this work, we choose gene scores as input because they are well defined and have a small feature space. Also, it is possible to further connect the model trained on gene scores to scRNA-seq models, and vice versa. There are different ways of summarizing gene scores[18,30] and our first question is how to utilize these gene score models. In total, ArchR provides 54 variations of gene score models (details in Supplementary Note 2), and its recommended one is shown to be the most accurate to infer gene expression in matched scATAC-seq and scRNA-seq data. From real data analysis, we show that using the ArchR recommended gene score model achieves good celltyping performances from Cellcano (details in Supplementary Note 3).

We next evaluate Cellcano with the recommended gene score or fixed-size 500-bp bin counts as input in both human PBMCs and mouse brain celltyping tasks. The comparison of prediction performances from human PBMCs is shown in Fig. 2a and Supplementary Fig. 1a, b. The two types of inputs produce comparable prediction accuracies in most celltyping tasks, while results in ARI and macroF1 show that using gene scores is significantly better. In mouse brain celltyping tasks, Cellcano with gene scores as input is better than using fixed-size bins in 62 out of 63 prediction results (Fig. 2b, Supplementary Fig. 1c, d), except one in mouse brain celltyping task using ARI as measurement (Supplementary Fig. 1c). Overall, these results demonstrate that using gene scores as inputs works better than using bin counts. In addition, the computational time for using gene scores as input is much shorter (Fig. 2c). Considering both computational and prediction performances, we decide to use the ArchR recommended gene scores as Cellcano's default input.

### Properties of Cellcano anchors

Cellcano selects anchor cells from the target dataset based on the prediction entropy from the first round (details in the Methods section) and uses them as reference to predict cell types for non-anchors in the second round. The number of anchors is specified by user as a cutoff for the quantiles of entropies. For example, when using 0.3 entropy quantile cutoff, 30% of the cells in the target dataset will be selected as anchor cells. As an exploration, we first compare the performance between anchors and non-anchors under different quantile cutoffs (0.1 to 0.6 with step size 0.1) in human PBMCs celltyping tasks and mouse brain celltyping tasks. Results (details in Supplementary Note 4) show that the final prediction performance depends on a balance between anchor numbers and anchor accuracy.

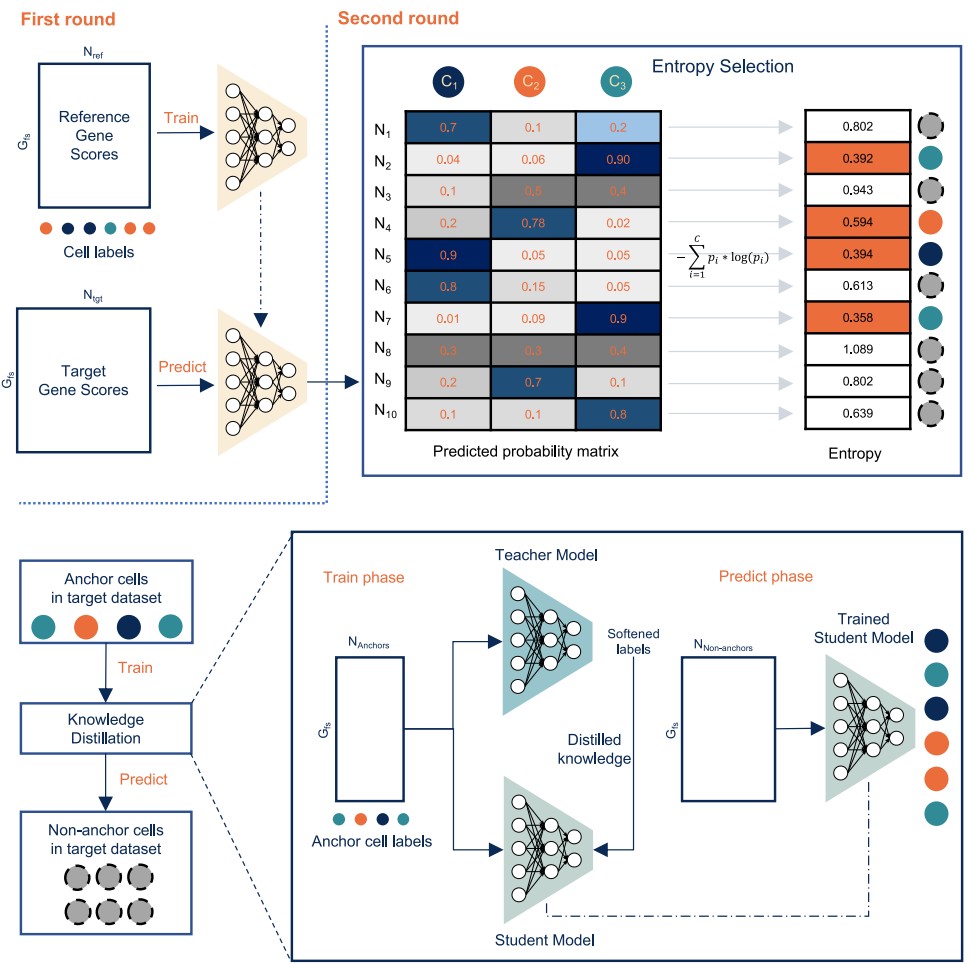

**Fig. 1 | Overview of Cellcano framework.** Cellcano adopts a two-round prediction strategy. In the first round, Cellcano trains a Multi-layer Preceptron (MLP) model on reference gene scores with known cell labels. Then, Cellcano uses the trained MLP to predict cell types on target gene scores. When the target size is sufficiently large, Cellcano starts the second round by selecting anchors. With the predicted probability matrix obtained from the first-round prediction, entropies are calculated for each cell. Cells with relatively low entropies are selected as anchors to train a Knowledge Distillation (KD) model. The trained KD model is used to predict cell types in remaining non-anchors.

We then summarize the final prediction performances using different entropy quantiles in human PBMCs celltyping tasks (Fig. 2d, Supplementary Fig. 2a, b) and mouse brain celltyping tasks (Fig. 2e, Supplementary Fig. 2c, d). Each celltyping task has a prediction baseline which is calculated as the average performance by using different quantile cutoffs. We calculate the gains/losses for using each quantile cutoff against the average performance. Overall, the performances are stable when using 0.2 or above as quantile cutoffs (the median Acc varies within −0.4% ~ +0.9% in human PBMCs celltyping tasks and −0.9% ~ +1.4% in mouse brain celltyping tasks). The worst performance occurs when using 0.1 as the quantile cutoff. This can be explained by the small training size in the second round and the failure of capturing the target data distribution. In conclusion, when using a moderate number of anchor cells, Cellcano can produce comparable prediction results. By default, we use 0.4 as the entropy quantile cutoff in our software implementation. Moreover, since Seurat also has an anchor selection step, we perform comparisons and show that Cellcano anchors are more accurate and can better capture the full scope of target data distribution (details in Supplementary Note 5).

### Cellcano outperforms existing supervised scATAC-seq celltyping methods

After deciding the input data and the anchor numbers for Cellcano, we compare Cellcano with other supervised scATAC-seq celltyping methods. We benchmark Cellcano against six competing supervised celltyping methods: Seurat[24], scJoint[25], Signac[26], EpiAnno[27], ACTINN[8], and SingleR[4]. Even though Seurat and scJoint are not specifically designed for scATAC-seq celltyping using scATAC-seq data as reference, they can take gene scores as input for cell type prediction. For Signac, we follow its recently published scATAC-seq integration vignettes to first process raw scATAC-seq data into peak counts and then perform data integration along with label transfer. For EpiAnno, we use ArchR to call peaks and count reads overlapping the peak regions to generate peak-by-cell matrices as its input. ACTINN is a deep learning based method which is very similar to the first-round prediction of Cellcano. SingleR is a correlation-based supervised scRNA-seq celltyping method. According to a recent survey study, SingleR is the second-best performer behind Seurat in scRNA-seq celltyping[12]. Even though ACTINN and SingleR are designed for scRNA-seq celltyping, they do not make any scRNA-seq specific assumptions on the input data and thus can take the gene scores as input for scATAC-seq celltyping. We include them because we want to explore whether existing scRNA-seq supervised celltyping methods can be directly applied to scATAC-seq with gene scores as input. In addition, we also include another set of comparisons by first removing the batch effect between reference and target datasets and then use an MLP to transfer cell labels (details in the next section). We put all the results together to make direct comparisons on prediction performances. We evaluate

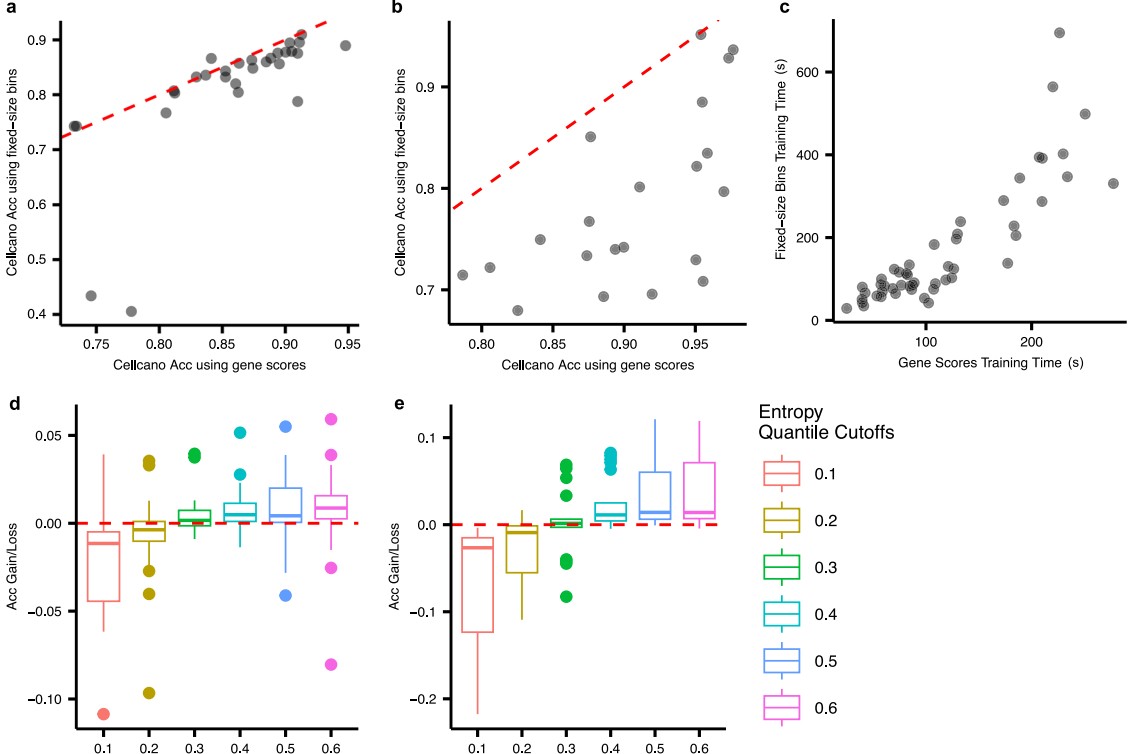

**Fig. 2 | Cellcano's parameter selection. a–c** Focus on exploring performances between using different input for Cellcano. **a, b** Accuracies comparison on Cellcano using genome-wide fixed-size bins and gene scores as input from (**a**) $n = 29$ human PBMCs celltyping tasks and (**b**) $n = 21$ mouse brain celltyping tasks. The red dotted lines are identity lines. **c** Model training time comparison using the two different inputs on all $n = 50$ celltyping tasks. **d, e** demonstrate the selection of the appropriate number of anchors. **d, e** Accuracy gains/losses using different entropy cutoffs on (**d**) $n = 29$ human PBMCs celltyping tasks and (**e**) $n = 21$ mouse brain celltyping tasks. Inside the boxes, the middle line indicates the median of the data while the bottom and upper lines indicate the 25th percentile and the 75th percentile of the data. Outside the boxes, the whiskers extend to the minimum and maximum values no greater than 1.5 times interquartile range. Those values outside the range are outliers, which are represented as dots with corresponding colors. Each box in (**d**) contains $n = 29$ prediction results and each box in (**e**) contains $n = 21$ prediction results. Source data are provided as a Source Data file.

the prediction performances from all methods by different metrics, including overall accuracy (Acc), adjusted rand index (ARI), macro F1 score (macroF1), Cohen's kappa (κ), median F1 score (medianF1), median precision, and median recall.

We first focus on the celltyping methods and compare the performances where we have one fixed gold standard target data (Fig. 3a, Supplementary Fig. 3). In total, there are seven celltyping tasks using different references. We order the boxplot according to the average performance. The results show that Cellcano achieves the highest average accuracy as 0.852 in the seven celltyping tasks (Fig. 3a), while scJoint is a close second with average accuracy as 0.837 and the third performer ACTINN has an average accuracy as 0.782. The accuracies from all other methods are significantly lower. For all other metrics (Supplementary Fig. 3), Cellcano and scJoint in general have the highest performances compared to all other methods, consistent with the results in prediction accuracy. Overall, the third best performer is ACTINN which is a variation of Cellcano first-round prediction. The performance differences between Cellcano and ACTINN indicate the performance improvements by introducing our second-round prediction.

We then evaluate the performances in all other 22 human PBMCs celltyping tasks (Fig. 3b, Supplementary Fig. 4). Since the celltyping tasks involve different target datasets, the baseline performance for each celltyping task can vary. We eliminate such baseline effect by computing the performance gains/losses for each method against the average. To be specific, we take the average of the prediction performances from all seven methods for each celltyping task, and then subtract the average from the performances for each method. From

these experimental scenarios, Cellcano ranks first in average accuracy gain and average ARI gain where Signac ranks the second. Signac slightly outperforms Cellcano in average macroF1 gain. Overall, ACTINN ranks the third. Similarly, we evaluate the performances in 21 mouse brain celltyping tasks (Fig. 3c, Supplementary Fig. 5) and observe that Cellcano again outperforms all other methods with most accuracy gain as 0.144. In the meantime, Signac acts as the second-best performer with accuracy gain as 0.134 and ACTINN acts as the third-best performer with accuracy gain as 0.120. Note that EpiAnno fails to generate results for two relatively larger (over 32k cells) celltyping tasks due to memory limit. Taking all 50 celltyping tasks together, we perform a paired $t$-test on Accuracy, ARI and macroF1 in three comparisons: (1) Cellcano and ACTINN, (2) Cellcano and scJoint, and (3) Cellcano and Signac. The test statistics show that Cellcano performs significantly better than ACTINN ($p$-value: 4.857e-3), scJoint ($p$-value: 1.645e-3) and Signac ($p$-value: 0.023) in Accuracy. Results hold for all comparisons in ARI. For macroF1, Cellcano slightly outperforms ACTINN while largely outperforms scJoint and Signac. In summary, Cellcano outperforms all other methods considering all scenarios: two systems (human PBMCs and mouse brain), 50 celltyping tasks, and seven metrics.

To further demonstrate how the two-round procedure in Cellcano outperforms, we use one celltyping task (one FACS-sorted human PBMCs dataset as target, a combination of four individuals from Satpathy et al.[33] PBMCs dataset as reference) as an example to visualize the prediction results after each round by tSNE and UMAP. Figure 3d and Supplementary Fig. 6a labels the ground truth cell types provided by FACS. After the first-round prediction, some cells in B cell and

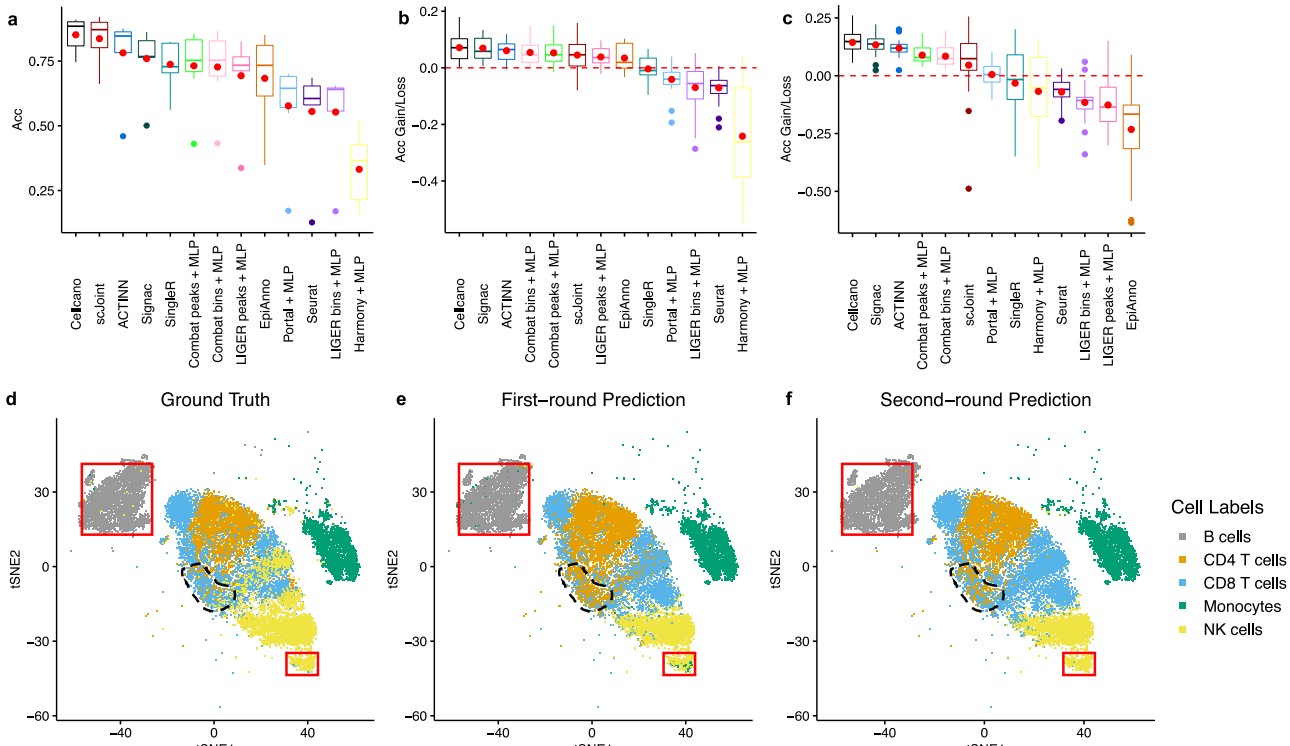

**Fig. 3 | Performance comparisons between Cellcano and other competing methods along with illustrations on how Cellcano outperforms. a–c** Accuracy comparisons between Cellcano, Seurat, scJoint, Signac, SingleR, ACTINN, and EpiAnno along with other integration with label transfer methods on (**a**) $n = 7$ celltyping tasks using one human PBMCs FACS-sorted dataset as target, (**b**) $n = 22$ more human PBMCs celltyping tasks and (**c**) $n = 21$ mouse brain celltyping tasks. Inside the boxes, the middle line indicates the median of the data while the bottom and upper lines indicate the 25th percentile and the 75th percentile of the data. Outside the boxes, the whiskers extend to the minimum and maximum values no greater than 1.5 times interquartile range. Those values outside the range are outliers, which are represented as dots with corresponding colors. Note that we use red dots to indicate the mean of the data. Each box in (**a**) contains $n = 7$ prediction results, each box in (**b**) contains $n = 22$ prediction results and each box in (**c**) contains $n = 21$ prediction results. The boxplots are ordered to have the leftmost method with the highest average performance. **d–f** t-SNE plots from one of the celltyping tasks using FACS-sorted dataset as target that contains $n = 21,214$ cells. The cells are colored with (**d**) ground truth labels; (**e**) Cellcano first-round predicted labels; and (**f**) Cellcano second-round predicted labels. The highlighted areas illustrate Cellcano's ability to correct wrongly assigned cells predicted from the first round. Source data are provided as a Source Data file.

natural killer (NK) are wrongly predicted as Monocytes (Fig. 3e and Supplementary Fig. 6b, red boxes). After the second round, the wrong predictions are corrected (Fig. 3f and Supplementary Fig. 6c, red boxes). Another observation is that many CD8 T cells on the boundary between CD4 T cell and CD8 T cell clusters (black dotted line area) are not correctly predicted. After the second round, most of these cells are correctly assigned back to CD8 T cells. We also visualize the predicted correctness, entropy, and predicted probabilities of CD8 T cells before and after second-round prediction (Supplementary Fig. 7a–f). The increased correctness and confidence in predicting cell types demonstrate the advantage of having our second-round prediction with KD model. Similarly in an example mouse brain celltyping task (Supplementary Fig. 8), some inhibitory neurons are wrongly predicted as Astrocytes and Microglias are wrongly predicted as Oligodendrocytes after the first-round prediction and those are corrected after the second-round prediction (Supplementary Fig. 8, red boxes). These visualization examples demonstrate the advantage of having our second-round prediction with KD model.

**Cellcano works better than prediction with batch effect removed**

A key advantage of the two-round approach in Cellcano is that training a model using anchors in target data alleviates the distributional shift problem between the reference and target data. The distributional shift is often caused by batch effect in high-throughput data. This leads to a question whether our two-round strategy is better than the one where we first remove batch effect and then apply a direct prediction.

According to a recent benchmark study[34], LIGER[35] and ComBat[36] are the top performers when integrating scATAC-seq datasets. Although we have proven that using gene scores as input is the best choice for Cellcano, in this benchmarking study, genome-wide bins or peaks are suggested as inputs for the integration tasks. We therefore follow the suggestions and include four top-performing integration combinations into our comparison: LIGER with genome-wide bins as input, LIGER with peaks as input, ComBat with genome-wide bins as input and ComBat with peaks as input as integration methods. We are also interested in knowing how batch-effect removed methods work with gene scores as input. Therefore, we added Harmony[37], which was demonstrated to have the best performance and shortest running time in previous batch-effect removal benchmark study in scRNA-seq data[38]. In the meantime, we also included Portal[39], a recently published integration method which has not been benchmarked and can take gene scores as input. After performing the integration between reference and target datasets, we apply MLP as classifier to transfer cell labels according to the integrated output (details in Methods section). We evaluate the prediction performances by Acc, ARI and macroF1.

As mentioned earlier, we put all prediction results from celltyping and integration with label transfer into boxplots (Fig. 3a–c, Supplementary Fig. 3a, b, Supplementary Fig. 4a, b, Supplementary Fig. 5a, b) for a direct comparison. Since boxplots provide marginal distributions which represent the overall performances, we add heatmaps (Supplementary Figs. 9, 10) with original prediction performances to show a full scope comparison. We categorize the heatmaps by different types of celltyping tasks and make the leftmost column have the

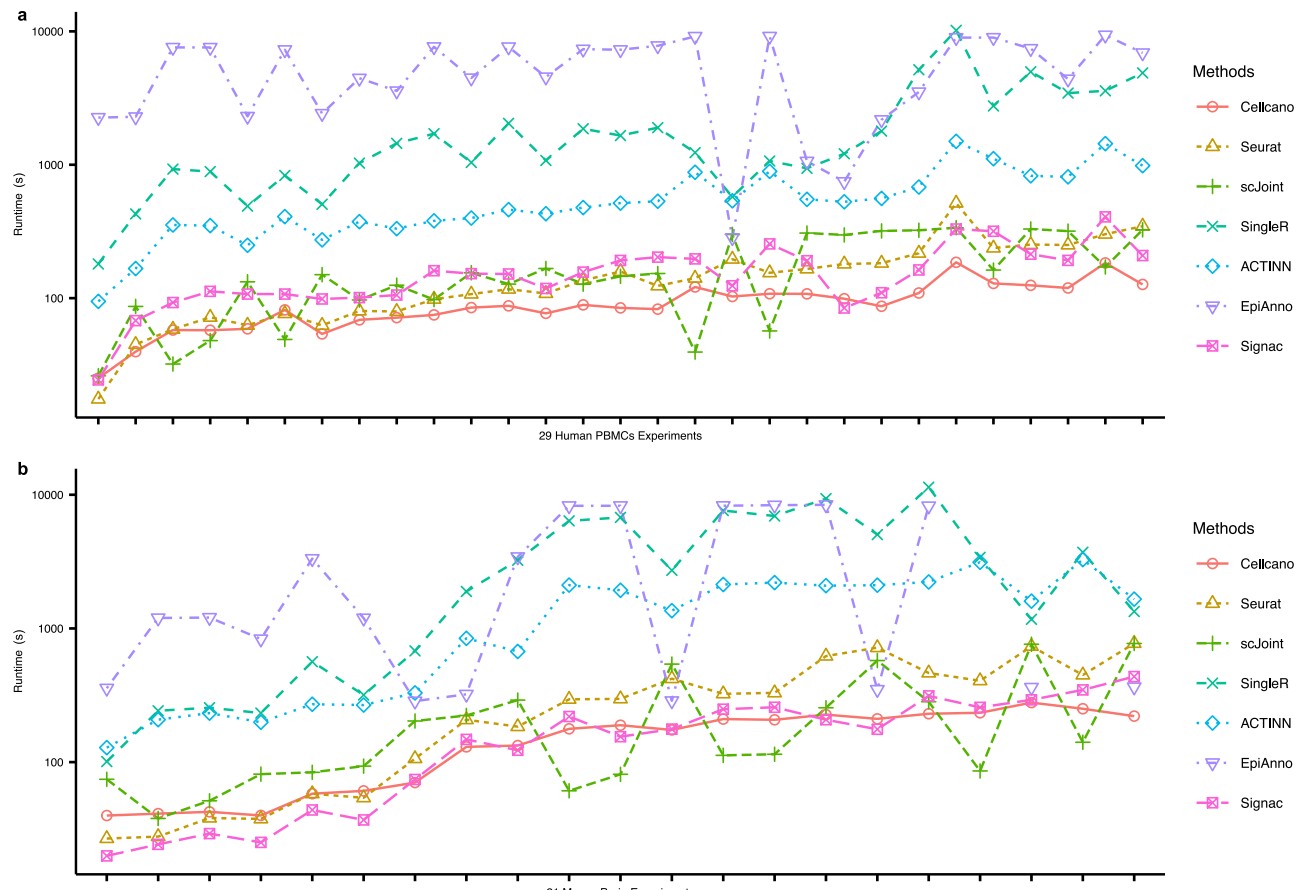

**Fig. 4 | Computational performance comparison. a**, **b** Run time comparisons among Cellcano, Seurat, scJoint, Signac, SingleR, ACTINN, and EpiAnno on (**a**) *n* = 29 human PBMCs celltyping tasks and (**b**) *n* = 21 mouse brain celltyping tasks. The *x*-axis indicates each celltyping task and is ordered by the total number of cells in the reference and target datasets. Note that EpiAnno fails to generate results for two mouse brain celltyping tasks where the cell numbers are large. Source data are provided as a Source Data file.

highest average performances. When focusing on all integration with label transfer methods, ComBat with peaks as input and ComBat with genome-wide bins as input rank top, however, they do not outperform the top celltyping performers and thus are inferior to Cellcano. We generate a low-dimensional visualization before (Supplementary Fig. 11a) and after the batch effect removal (Supplementary Fig. 11b–e) on one example where one FACS-sorted PBMCs data is taken as target and four individuals from Satpathy et al. are combined as reference. We can observe that even when the batch effect removal methods work well on integrating reference and target datasets or integrating individuals (Supplementary Fig. 11c, d using LIGER and Portal), the celltyping results are not necessarily better. In conclusion, these comparisons demonstrate that Cellcano can handle data from different individuals and batches in both reference and target data. Cellcano does not need to remove batch effect and steadily outperforms other integration with label transfer methods. This provides the possibility of training predication models using a large compendium of datasets.

**Cellcano is computationally efficient and scalable**
We evaluate the computational performance of Cellcano and show all celltyping methods' runtime for all celltyping tasks (Fig. 4a, b). For fair comparisons, we combine the training time and prediction time into an overall runtime for Cellcano and EpiAnno. This is because all other methods need both reference and target datasets as input to do prediction. Here, we do not consider the data pre-processing time (such as the time used for generating peak counts or gene scores from the raw data). We sort the celltyping tasks by the total number of cells in the

reference and target datasets. The results indicate that when the cell number is low, Cellcano, Seurat and scJoint use about the same runtime. However, when the cell number starts increasing, Seurat and scJoint can be three times slower than Cellcano. Signac is 2 ~ 3 slower than Cellcano when predicting cell types for human PBMCs tasks while its running time is comparable to Cellcano in mouse brain celltyping tasks. All other methods are 5 ~ 100 times slower than Cellcano. The reason why ACTINN as one-round prediction is slower than Cellcano is because ACTINN uses all genes for training while Cellcano selects 3000 genes as features. An additional advantage is that Cellcano is a supervised celltyping method, the pretrained models can be re-used in future predictions, which means the runtime can be further reduced with the first-round pretrained model as input.

## Discussion
Computational celltyping for single cell omics data is an important problem. Such methods are under-developed for scATAC-seq data. In this work, we develop Cellcano, a two-round supervised scATAC-seq celltyping method. Due to distributional shift, the first-round prediction can be inaccurate, and the anchors can be noisy. The KD model in the second round is thus used to distill the knowledge from a noisily labeled input. We have shown in 50 celltyping tasks with data from two systems (human PBMCs and mouse brain) that Cellcano significantly outperforms other celltyping methods and integration with label transfer methods both in prediction and computational performances. Cellcano is also robust against the anchor selection procedure and batch effects in the data.

Cellcano has several advantages and methodological features. First, Cellcano uses gene scores as input, which has many advantages compared to using bin or peak counts: (1) genes have a much smaller feature space, which significantly improve the computational performance; (2) genes are shared among datasets, which provides potential to be further connected to other modalities, such as gene expression data. We show that using gene scores works as equally well or even better than using bin counts as input. Secondly, Cellcano implements strategies in selecting and using anchors. The MLP in Cellcano can better capture the non-linear relationship between the gene scores and the corresponding cell types. In addition, the KD model is robust to anchors with noisy labels. Moreover, Cellcano does not need to jointly operate on the reference and target datasets, like Seurat, Signac, and scJoint does. This allows Cellcano to be trained on a compendium of reference datasets and provide a pre-trained model.

There are some further developments for Cellcano we plan to work on. First, Cellcano can be adapted to other celltyping scenarios, for example, cross-modality predictions (using scRNA-seq as reference for scATAC-seq celltyping), celltyping in single-cell DNA methylation, etc. Another interesting question is to use multimodal reference data, for example, to jointly use scRNA- and scATAC-seq data as reference to improve celltyping results for either scRNA- and scATAC-seq data. Such an approach can potentially further improve prediction performance.

## Methods

### Overall scheme of data processing and analysis

Conceptually, the design for the celltyping task is rather straightforward (Supplementary Fig. 12). Below are the details in each step.

1. Data collection and genome unification: We collected four datasets from human PBMCs and two datasets from mouse brains and then downloaded the raw scATAC-seq data (10X fragment files or bam files). The original raw data sometimes are aligned to different versions of genomes, we therefore use liftOver to unify the human PBMCs data to hg19 genome and the mouse brains data to mm10 genome. After unification, we use ArchR to process the raw scATAC-seq data. Note that for Signac, we need an additional step to convert bam files into fragment files with sinto (https://timoast.github.io/sinto/installation.html).

2. Preprocess raw data with ArchR: In ArchR, we set genome hg19 for human PBMCs datasets and mm10 for mouse brain datasets. Then, we load the downloaded fragment files or bam files as input for ArchR to generate the *ArrowFiles* with *createArrowFiles*() function. In the function, two parameters serve with quality control purpose: *minTSS* and *minFrags*. We adjust the thresholds according to original papers to obtain high-quality cells and we use default thresholds from ArchR for those datasets with no explicit quality control information provided.

3. Generate feature matrices: The gene score matrices and genome-wide fixed-size bin counts are generated using the default setting in ArchR. The gene score matrix is generated with ArchR recommended gene score model (details in Supplementary Note 1). The bin counts are generated with 500-bp bins genome-wide. This results in around 6 million bins in hg19 and 5 million bins in mm10. To accelerate the data loading time, we filter out the bins with non-zero counts in less than 1% cells to reduce the feature space. The peak-by-cell matrices generation needs additional peak calling steps in ArchR. To reuse the *ArrowFiles* generated earlier, we put *ArrowFiles* from all human PBMCs datasets together and call peaks. ArchR first clusters cells and then creates pseudo-bulk replicates to assure the reproducibility of peak calling. Once the peaks are obtained, reads are counted on the peak regions to generate the peak count matrices. The same procedure has been performed in mouse brain datasets. Note that for Signac, we use raw fragment files as input.

4. Cell type curation and construct celltyping tasks: Once having all generated feature-level matrices, we curate the cell types for human PBMCs datasets and mouse brain datasets (details in Supplementary Note 6). Then, according to our prediction task designs on celltyping (details in Supplementary Note 1; celltyping tasks listed in Supplementary Dataset 1 and 2), we select one reference dataset with cell type information and one target dataset without cell type information to perform Cellcano as well as other benchmarked supervised celltyping methods. As for the batch effect removal methods, we first integrate the reference and target datasets without cell type information. Then, when we have the corrected data, we extract the corrected reference data and feed in the cell type information. Finally, we use MLP to predict cell types in the corrected target data.

### Input data for Cellcano

Cellcano is a supervised celltyping model, therefore, cell type information of the reference dataset is required. As for the input data format, Cellcano can use either raw scATAC-seq data or processed gene score matrices as inputs. As mentioned in the last section, if the input is raw scATAC-seq data, ArchR is first performed to generate gene scores for the reference and target datasets. If users already have derived gene scores from the reference data, they can also be taken as the input for Cellcano. In such case, it is recommended to have the same gene score calculation procedures for reference and target datasets to assure the first-round prediction performance.

### Cellcano model

Once we have the gene scores from both reference and target datasets, we assume there are $G$ genes and $N$ cells in the reference, and $M$ cells in the target data, we define the gene score matrices in reference and target data as $X_{ref} \in \mathbb{R}^{G \times N}$ and $X_{tgt} \in \mathbb{R}^{G \times M}$, respectively. In the reference gene scores, we first perform a feature selection step to select representative features. The features are selected by F-test with known cell type labels, represented as $C_{ref} \in \mathbb{R}^{N \times 1}$. We have previously shown that features selected by F-test in reference data can provide the best results in supervised scRNA-seq celltyping[13]. By default, we select top 3000 genes with the largest F-statistics. We obtain the reference and target gene scores for the selected features and perform data normalization. To be specific, we normalize the cell-wise gene scores so that the total gene scores sum to 10,000 for each cell. We then take log-transformation on the normalized gene scores plus 1. After that, we perform gene-wise standardization on the log normalized gene scores so that each gene will have zero-mean and unit-variance. The standardization is a recommended procedure for performing efficient backpropagation in neural networks[40].

In Cellcano's first-round prediction, we first train an MLP model with a *ReLU* activation function to capture the non-linear mapping between the $X_{ref}$ and $C_{ref}$. For a multi-class classification with $K$ cell types, the cell type label $C_{ref}$ is one-hot encoded to a binary matrix with dimension $N \times K$. The one-hot encoding labels the corresponding class as 1 and all others as 0 for each cell. The last layer of MLP is connected to a softmax function to convert the outputs from the last layer of the MLP to probabilities. The softmax function is represented by

$$\sigma(Z_i) = \frac{\exp\left(\frac{Z_i}{T}\right)}{\sum_{k=1}^{K} \exp\left(\frac{Z_k}{T}\right)}. \tag{1}$$

Here, $Z_i$ represents the outputs from the last layer of the MLP, and $T$ is a hyperparameter representing the temperature of the softmax function. The larger the $T$ is, the smoother the $\sigma(Z_i)$ will be. We set $T = 1$ in the first-round MLP model. During training, we use cross-entropy as the loss function to minimize the distributional difference between the one-hot encoded cell type label $p$ and the predicted cell type

probabilities $\sigma(Z)$:

$$H(p, \sigma(Z))) = -\sum_{i=1}^{N}\sum_{k=1}^{K} p_{ik} \log(\sigma(Z_i)_k). \qquad (2)$$

After training the MLP model, we apply the trained MLP model to the target data to obtain the probabilities for each cell being in each cell type.

When the target data size is small, Cellcano takes the class with the largest probability as the final predicted cell type for each cell and stops. When the target size is large (over 1000 cells by default), we perform a second-round prediction. We first select anchors from the target, and we aim at selecting accurate anchors, which can also capture the full scope of target distribution to guide the second-round prediction. With the first-round predicted probabilities, denoted as $q_{ik}$ for cell $i$ being in cell type $k$, we calculate the entropy $E^{M \times 1}$ for all M cells as

$$E_i = -\sum_{k=1}^{K} q_{ik} \log(q_{ik}). \qquad (3)$$

When a cell label is more confidently assigned, its entropy over the predicted probabilities is lower, and the prediction is in general more accurate (Fig. 4a, Supplementary Fig. 7–8). Once we have entropies for all cells, we select 40% cells with the lowest entropies as anchors for each cell type to form the new reference dataset for second-round training. Sometimes the cell type composition in anchors can be very skewed, which could affect the performance when training the second-round model. Therefore, we first calculate the average number of cells per cell type according to the predicted cell types in the anchors, and then oversample the cell types with fewer cells to the average number. This can ensure that there are enough training cells in all cell types in the anchor. Since some anchors will be mistakenly predicted, we apply the KD model in the second-round training to deal with the issue, detailed in next section. The model trained in the second round will be used to predict cell types for non-anchors. Finally, we combine the cell types predicted for the anchors (from the first round) and non-anchors (from the second round) as our final cell type calls.

**The Knowledge Distiller (KD) model.** Although the anchors cannot be perfectly predicted from the first round, they are important complementary training data for improving prediction, since these cells are from the exact same target domain where we previously lack supervision. To deal with training data with noisy labels, we implement a self-Knowledge Distiller (KD) model in the second-round training. The KD technique was originally proposed to transfer the knowledge learned from a sophisticated teacher model to a light-weighted student model, by treating the prediction results produced from the teach model as the "soft labels" for training the student model[41]. Inspired by this and several recent works[29,42], we propose to use the teacher-student interaction to alleviate the noisy label problem. Specifically, the teacher model distills knowledge from both clean supervision and noisy supervision by producing "soft labels" as the training targets of the student model. Compared to the "hard labels" that only contain overconfident 1's and 0's, "soft labels" are smoothed and thus more noise-tolerated[43]. Also, there are cell types sharing similar profiles during celltyping which fits the fine-grained classification setting in the KD model. In Cellcano, we apply a "self-KD model" where we have the exact same structure for the teacher model and the student model. We set them to be vanilla MLPs of two hidden layers with 64 and 16 nodes, respectively. To let the model be more generalizable, we put the dropout layer right after the input layer. We use *ReLU* as the activation function.

We first train the teacher model with the anchors as input. To make the label "softer", we set the temperature $T$ of the softmax function to be larger. We use cross-entropy loss for the teacher model, then train the student model with the teacher's "soft labels" as well as the one-hot encoded "hard labels". The idea is to learn a label smoothing regularization so that the label distribution can be better captured. The KD loss function for the student model is a weighted average of two losses, which is shown in the equation below:

$$L_{KD} = \alpha H\left(p, q_s^{T_1}\right) + (1-\alpha)KL\left(q_t^{T_2}, q_s^{T_2}\right). \qquad (4)$$

Here, $T_1$ and $T_2$ are temperatures in the softmax functions, and $\alpha$ is a hyperparameter for balancing the two losses. The first part of the KD loss is a cross-entropy loss where the student prediction $q_s$ is guided by "hard labels" (anchor cell types from first-round prediction), and we set the $T_1$ as 1. The second part represents the Kullback–Leibler (KL) divergence loss which measures the probability distribution distances between the soft teacher prediction $q_t$ and the soft student prediction $q_s$, where $T_2$ can be adjusted. We set $T_2 = 3$ for the second part to soften the label distribution. Overall, we set $\alpha$ as 0.1 to value more on the teacher model's "soft labels". The KD model is trained for 30 epochs.

## Supervised celltyping methods

We benchmark Cellcano to six competing methods: Seurat, scJoint, Signac, EpiAnno, ACTINN, and SingleR. For all methods except EpiAnno and Signac, we use the scATAC-seq gene score matrix before feature selection in the place of scRNA-seq gene expression matrix as the reference and follow their default procedures for celltyping. In Seurat, we choose reciprocal principal component analysis (RPCA) to calculate the joint embedding of reference and target datasets as it is proved to have better integration performance in a benchmark paper[34]. For Signac, we use raw fragment files as input and follow the vignettes provided by Signac on scATAC-seq data integration. We notice that overlapping genomics regions to obtain peak count matrices for target datasets takes a much longer time than generating gene scores with ArchR. For EpiAnno, we use peak counts matrices as input. To accommodate the memory limitation of EpiAnno, we set the hyper parameter *peak rate* as 0.08 or 0.05 for large input matrices, while keeping the original 0.03 *peak rate* for remaining matrices.

## Integration with label transfer methods

We benchmark Cellcano to six combinations of batch-effect removal methods with different feature-level inputs: LIGER with peaks as input, LIGER with genome-wide bins as input, ComBat with peaks as input, ComBat with genome-wide bins as input, Harmony with gene scores as input and Portal with gene scores as input. We use corresponding feature matrices derived from ArchR as input for each combination and use the default procedures to remove batch effect between the reference and target datasets. ComBat returns corrected counts which has the same dimension as the original data while LIGER, Harmony, and Portal return joint embeddings with size as 20, 50, and 20, respectively. We then split the corrected counts or joint embeddings into reference and target datasets according to the batch information before integration and perform celltyping with MLP. Here, the MLP model has the same structure and settings with the MLP model in Cellcano's first-round prediction.

## Evaluation Metrics

We choose overall accuracy, ARI and macroF1 for performance evaluation. Accuracy describes the number of correctly assigned cells divided by total number of cells. This can be used as an indicator to show how well most cells are assigned. ARI measures the cluster concordance between the true labels and predicted labels. MacroF1 treats all cell types equally, which puts more emphasis on the accuracies of smaller clusters compared to other metrics. For a fair comparison, we

also include median F1 score (medianF1), median precision, median recall, and Cohen's kappa (κ), which were used in the EpiAnno paper. The medianF1, median prediction and median recall regard predicting each cell type as a binary classification task and calculate the median performance for each cell type. Cohen's kappa measures the agreement between labels from the ground truth and the predictor. In summary, these metrics measure different perspectives and can be used to fairly reflect prediction performance among different performers.

## Statistics and reproducibility

In this work, we set random seed as 2022 for Python random package along with tensorflow package to ensure the reproducibility of our results. No statistical method was used to predetermine sample size. Cells with low-quality were excluded based on standard scATAC-seq preprocessing procedures. The experiments were not randomized. The Investigators were not blinded to allocation during experiments and outcome assessment.

## Reporting summary

Further information on research design is available in the Nature Portfolio Reporting Summary linked to this article.

## Data availability

All datasets are publicly available, and the access numbers or the downloaded websites are provided by the original publications. The Satpathy et al.[3] data used in this study are available in the Gene Expression Omnibus (GEO) dataset under accession code "GSE129785". The Granja et al.[44] data is available in the GEO dataset under accession code "GSE139369". The 10X PBMCs data is available in the 10X genomics datasets. We downloaded the raw data under the Single Cell Multiome ATAC + Gene Expression category named PBMC from a Healthy Donor – Granulocytes Removed Through Cell Sorting (10 K) processed by Cell Ranger ARC 2.0.0 (https://www.10xgenomics.com/resources/datasets/pbmc-from-a-healthy-donor-granulocytes-removed-through-cell-sorting-10-k-1-standard-2-0-0). The FACS PBMCs[45] data is available in the GEO dataset under accession code "GSE123578". The Lareau et al.[45] data is available in the GEO dataset under accession code "GSE123581". The Cusanovich et al.[46] data is available in the Mouse Atlas dataset under Downloads tab (https://atlas.gs.washington.edu/mouse-atac/data/). The liftOver chain files to convert human genome build hg38 to hg19 data are available in the UCSC file server saved as hg38ToHg19.over.chain.gz (https://hgdownload.cse.ucsc.edu/goldenpath/hg38/liftOver/) and the chain files to convert mouse genome build mm9 to mm10 data is available in the USCS file server saved as mm9ToMm10.over.chain.gz (https://hgdownload.cse.ucsc.edu/goldenpath/mm9/liftOver/). More information on the datasets can be found in Supplementary Table 1 and details on data preprocessing are provided in Supplementary Note 2. All other relevant data supporting the key findings of this study are available within the article and its Supplementary Information files or from the corresponding author upon reasonable request. The results of celltyping prediction tasks along with supporting visualization information that appear in all figure panels are provided in the Source Data files. Source data are provided with this paper.

## Code availability

Cellcano code is freely available on GitHub (https://github.com/marvinquiet/Cellcano) and Zenodo (https://doi.org/10.5281/zenodo.7686209)[47]. The software package has been released both on PyPI (https://pypi.org/project/Cellcano/) and Anaconda (https://anaconda.org/marvinquiet/cellcano-all). Users can choose either option to easily install our package. Detailed tutorials on installation and usage are also provided (https://marvinquiet.github.io/Cellcano/). Cellcano python package is built upon Python and the recommended version is v3.8. There are several Python package dependencies, including tensorflow (v2.7.1), anndata (v0.7.4), scanpy (v1.8.2), numpy (v1.19.2), h5py (v2.10.0), keras and rpy2. R environment and the R package ArchR (v1.0.1) are also suggested to be installed for generating gene scores.

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

## Acknowledgements

This work is partially supported by the NIH award R01GM122083 for HW and WM. HW was also partially supported by the Strategic Priority Research Program of Chinese Academy of Sciences, Grant No. XDB38050100.

## Author contributions

H.W. conceived the study. W.M. designed and implemented the model. W.M. developed the Cellcano software and analyzed the results. J.L. provided advice on algorithms and helped on benchmarking and testing the software. W.M. and H.W. wrote the paper with input from J.L.

## Competing interests

The authors declare no competing interests.
