## [Peer Review File · Nature Communications]

Cellcano: supervised cell type identification for single cell ATAC-seq dataREVIEWER COMMENTS

Reviewer #1 (Remarks to the Author):

The authors describe Cellcano as a method for performing celltyping of ATAC datasets using an annotated reference ATAC dataset. The method itself is more of an integration approach and not necessarily celltyping, as the cell type is determined by label transfer after integration. Overall the text is long and needs to be heavily restructured. The comparison to RNA-based methods is not appropriate in the way that they are performed where ATAC gene scores are used instead of RNA. The presentation of the figures is confusing and needs to be reworked as well.

The intro is too long, there is too much text summarizing standardized workflows for scATAC analysis that are not directly relevant to the method being described. Much of this should be trimmed. The description of the methods that use scRNA to annotate cell types in scATAC could also be shortened to a single sentence with appropriate references. The intro should be able to be tightened up and shortened to less than half the current length.

Why is it called Cellcano? It does not seem to be based in an acronym and has no reference to celltyping or ATAC.

More detail should be provided on how parameters are determined. E.g. "The anchors are defined as the ones with higher confidence in first-round cell type prediction." Is unclear. Is this the set of anchors chosen out of all possible cell-cell links? What determines higher confidence? Entropy is alluded to later – is that what is used? The description of the method needs to be restructured to be more clear and precise along with specifics on what metrics are used and the assumptions that are made.

What are the requirements for the 'reference' dataset? All that is detailed is that the gene activity scores of a reference dataset are used.

The comparison of methods that use RNA as a basis for annotation should be used with RNA datasets. Using gene scores from ATAC data is not what those tools are designed for and therefore the comparison is not appropriate.

When one has an ideal reference dataset – ie the same tissue and modality (ATAC) – then the standard method for celltyping is to simply integrate the datasets and perform label transfer. This would be the appropriate method to compare against, since it is essentially what Cellcano is doing. (NOTE the authors address this later – which should be earlier and used in the overall comparisons – though again, the main set of comparisons are not appropriate)

Throughout the authors use very vague qualitative terms with no supporting statistics in the text. When saying a method performs "better" provide quantitative information.

Claiming 50 experiments is misleading – the authors devised 50 means of assessing data from four PBMC and two mouse brain datasets.

The section on detailing why gene scores are used follows the overall performance analysis. This should precede the overall performance – same goes for the cellcano anchors section. The text in these could also be trimmed substantially.

I would suggest adding a paragraph each for the gene score choice and anchor properties within the cellcano framework section, summarizing the main points along with further detail in the methods section. Many details would make more sense there than in the main text.

Notably these sections have better reporting of quantitative metrics.

The "motivation for cellcano" section in the methods can be removed – this is just rehashing the intro and is not relevant to the methods.

The authors use tSNE, but often UMAP performs better on many datasets. It would be good to show both.

The Figure 2 panels D, E and F are hard to interpret. Cells are colored by ground truth, so how are they showing the prediction?

In Fig 3 there is a lot of variability in model performance across the four datasets. Why?

There needs to be a figure diagramming what the experiments are for the comparisons. A schematic of datasets used, how they are processed, and the specific tests would help greatly.

There are no plots showing the visualized brain data. Also the PBMC data shows very poor cluster separation...

Figures need to be provided showing the harmony integration and how that method performs, as it is the most relevant. Essentially cellcano is just an integration method and not a celltyping method. It integrates then performs label transfer, but the integration is where all of the relevance is.

Reviewer #2 (Remarks to the Author):

PRESENTATION OF THE PAPER:

In this paper, the authors present a supervised approach to catalog cells when only scATAC-seq data is available. The model uses existing pipelines (ArchR) to compute gene level statistics (gene scores). Then, it selects features through a statistical test and train a MLP. If the target set is big enough, cellcano performs a second round of annotations in which it selects anchors (based on an entropy determination of the confidence of the labels) and trains a knowledge-distiller model. To assess the confidence and accuracy of the proposed approach, the authors benchmark cellcano against 6 competing approaches, used 7 metrics and plenty of data sets.

One of the weak points of the approach is the calculation of gene scores for cell annotation. The authors acknowledge this point, compare against different methods that summarize gene information and demonstrate that theirs is a good choice. Additionally, for the second round of annotations, cellcano uses anchors selected on confidence of annotation of the training dataset.

The method is available online and detail documentation on how to install it and how to run is also available.

More generally, the approach is simple and uses a common algorithm, MLP, to perform the annotation. Given the results, the method is sound and seems to be a top-performer. However, there are many approaches performing the same task already so it lacks novelty, this is also evident by the many approaches they compared against. Using a KD model to address the distributional-shift is a clever and novel approach to address one of the fundamental problems in single-cell genomics, batch and condition correction. Due to this fact, I believe the method merits publication. In addition, the paper is well written and does not need major revisions.

Having said this, it will be great if the authors describe in more detail their intuition (hopefully supplemented by any mathematical proof) about the following topics (it can be done in the response letter).

1) Why does Cellcano and sc-joint perform similarly when they are based on completely different approaches ?

2) Why does cellcano needs 2 rounds of predictions? Why is an MLP not enough to perform annotations in a first pass? If the intuition is that cells are incorrectly labeled at the decision boundary, can you plot the label uncertainty next to the annotations before and after KD?

3) Can you describe if there could exist incorrect anchors with high confidence that would produce incorrect results? Can this happen if the number of cells in target and reference data set have completely different cell numbers?

Minor: This sentence does not make sense. Variational inference is not used as a purpose, it is a method to perform approximated posterior inference.

"ScANVI is a semi-supervised learning method which uses deep generative model for variational inference purpose to first integrate scRNA-seq datasets and then transfer annotations"

Introduction

We thank the editor and reviewers for their thorough reviews, thoughtful comments, and constructive suggestions for our manuscript. We appreciate the positive comments such as “sound”, “top performer”. Based on the editor and reviewers’ comments, we have carefully revised the manuscript (modifications in the manuscript are highlighted in red font). In this revision, we made following major modifications:

1. We have extensively modified the manuscript according to the comments of reviewer 1:
 - a. We significantly shorten the Introduction section.
 - b. We modify the structure of the Results section.
 - c. We provide more methodological details. In particular, we emphasized that Cellcano is not an integration method (as reviewer 1 commented) but a celltyping method which only focused on how to better predict cell types in the target dataset. We also stressed the underlying method assumption of Cellcano.
2. We have extensively revisited the benchmarking and added a whole new section on comparing Cellcano to other integration methods with label transfer.
3. We investigated and explained that the reasons for the variability in model performance from the same system is due to the differences between reference and target datasets.
4. We have addressed all comments from the reviewers and extensively revised the manuscripts according to reviewers’ comments. We also have added new figures and supplementary information to support our findings.

We addressed the individual points raised by the reviewers, as detailed below. The reviewers’ comments are in *italic blue*, and our replies are in regular font.

Response to Reviewer 1’s comments:

The authors describe Cellcano as a method for performing celltyping of ATAC datasets using an annotated reference ATAC dataset. The method itself is more of an integration approach and not necessarily celltyping, as the cell type is determined by label transfer after integration. Overall the text is long and needs to be heavily restructured. The comparison to RNA-based methods is not appropriate in the way that they are performed where ATAC gene scores are used instead of RNA. The presentation of the figures is confusing and needs to be reworked as well.

We thank the reviewer for the comments. We would like to first emphasize that Cellcano is not an integration approach. Our method does not try to remove the distribution discrepancies between reference and target datasets, and it does not transform either the reference or target data. Instead, our main purpose is celltyping and we do it without integrating the data. The essence of our method is to directly train a classifier from part of the target data so that the classifier can capture target data distribution. Since the target dataset does not have cell type information, we first use the classifier trained on the reference data to predict cell types on the target datasets. Then we choose cells whose cell types are relatively better predicted (by looking at their prediction entropy) as anchors and consider them as the “true” cell type labels for the target datasets. We have shown that the accuracies of anchors are usually high and reliable. To handle the inaccurately predicted cell labels in the target data, we apply the knowledge distiller (KD) model to smooth the label distribution and further guide the classification in the non-anchor cells.

For the suggestion on shortening the text: we have extensively restructured and modified the manuscript. In particular, we heavily trimmed the Introduction section, the subsection “*The choice of using gene scores as input*” in the Results section, and the subsection “*Properties of Cellcano anchors*” in the Result section. The trimmed version is only about half of the original length.

For suggestions on wordings, figures, and benchmarking study construction, we respond in the following sections respectively.

(Comment 1) The intro is too long, there is too much text summarizing standardized workflows for scATAC analysis that are not directly relevant to the method being described. Much of this should be trimmed. The description of the methods that use scRNA to annotate cell types in scATAC could also be shortened to a single sentence with appropriate references. The intro should be able to be tightened up and shortened to less than half the current length.

Thank you for your suggestions. We have heavily trimmed our introduction part by removing nearly half of the content. The modification is based on the following suggestions:

- Use one or two sentences to introduce experimental methods in scATAC-seq celltyping.
- Remove the scATAC-seq data analysis pipelines and focus more on introducing related methods as well as their limitations.

(Comment 2) Why is it called Cellcano? It does not seem to be based in an acronym and has no reference to celltyping or ATAC.

The method name is not an acronym. We designed a logo for our software, shown in the software page at <https://marvinquiet.github.io/Cellcano/>. The logo looks like an erupting volcano, where the shape of the volcano mimics the structure of a multiple layer perceptron (MLP, which is used as classifier in our work), and the cell types are erupted out of it. Thus, we call the method and the software Cellcano.

(Comment 3) More detail should be provided on how parameters are determined. E.g. “The anchors are defined as the ones with higher confidence in first-round cell type prediction.” Is unclear. Is this the set of anchors chosen out of all possible cell-cell links? What determines higher confidence? Entropy is alluded to later – is that what is used? The description of the method needs to be restructured to be more clear and precise along with specifics on what metrics are used and the assumptions that are made.

Yes, the anchors are chosen as the cells with low entropies computed from the first-round prediction probabilities. The assumption is that if a cell has low entropy from first-round prediction, it is more likely to be accurately predicted. We demonstrated that this assumption is true in real data explorations shown in Supplementary Figure S14-15. More in-depth analyses can be found in Supplementary Note 3.

We have rephrased the last paragraph in the Introduction section as well as the subsection “*The Cellcano framework*” in the Results section by emphasizing the assumption and the procedure for entropy calculation.

(Comment 4) What are the requirements for the ‘reference’ dataset? All that is detailed is that the gene activity scores of a reference dataset are used.

The requirement for the reference dataset is that it must have cell type labels. We have added a new subsection “*Input data for Cellcano*” under the Methods section to provide more details on these requirements for the reference dataset.

As for the reference data format, it can either be raw data or processed gene scores. The raw data are either fragment files provided by 10X Genomics or bam files obtained from sequence alignment tools. When the raw data is given, Cellcano will first use ArchR to summarize it into gene scores. If users already have derived gene scores from the reference data, they can also be taken as the input for Cellcano. Note that it is recommended to have the same gene score calculation procedures for reference and target datasets to assure the first-round prediction performance. Details about the data preprocessing and analysis can be found in the newly added subsection “*Overall scheme of data processing and analysis*” under the Methods section.

(Comment 5) The comparison of methods that use RNA as a basis for annotation should be used with RNA datasets. Using gene scores from ATAC data is not what those tools are designed for and therefore the comparison is not appropriate.

Thanks for the comment. Even though we agree that the scRNA-seq celltyping methods are not designed for scATAC-seq, we would argue that some “*methods that use RNA as a basis for annotation*” can also be used in gene scores. Of course, if a scRNA-seq celltyping method makes assumptions specific for scRNA-seq data, it cannot be used in other data modalities. However, many of the scRNA-seq celltyping methods do not have scRNA-seq specific assumptions on the input data, for example, SingleR is based on correlation and ACTINN is based on a vanilla MLP. For these methods, the scATAC-seq gene scores (even though they have different distributions as gene expression from scRNA-seq) can certainly be used as input for celltyping. On the other hand, we found scANVI made a zero-inflated negative binomial assumption on the input data, thus it is indeed inappropriate to apply it on the scATAC-seq gene scores. We have removed scANVI from the comparison in this revision.

Another reason why we compare to those scRNA-seq methods is that the supervised celltyping method for scATAC-seq is limited. As of now, only EpiAnno is developed for supervised celltyping in scATAC-seq using scATAC-seq data as reference. Besides EpiAnno, only a few methods (Seurat and scJoint) use scRNA-seq data as reference and transfer cell labels to scATAC-seq datasets. To make the comparisons more comprehensive, we included some scRNA-seq methods. In fact, some of the scRNA-seq methods perform well, such as ACTINN.

We have added some discussion on this point in the subsection “*Cellcano outperforms existing supervised scATAC-seq celltyping methods*” under the Results section. We have also removed scANVI in all comparisons.

(Comment 6) When one has an ideal reference dataset – ie the same tissue and modality (ATAC) – then the standard method for celltyping is to simply integrate the datasets and perform label transfer. This would be the appropriate method to compare against, since it is essentially what Cellcano is doing. (NOTE the authors address this later – which should be earlier and used in the overall comparisons – though again, the main set of comparisons are not appropriate)

We first want to emphasize again that Cellcano does not do data integration. Moreover, some other scATAC-seq celltyping methods such as EpiAnno also does not include a data integration

step. Even though Seurat has a data integration function, it is designed for visualization purpose and not required or recommended for celltyping. Thus, we believe data integration is not an essential step for celltyping. In fact, our previous benchmark study for scRNA-seq supervised celltyping (Ma et al. 2021) demonstrated that data integration does not improve the prediction performances.

Intuitively, the data discrepancies between scATAC-seq datasets will be different from scRNA-seq due to the intrinsic characteristics of scATAC-seq data, therefore, a data integration step could potentially improve the celltyping results and that is the reason why we originally included the comparison to Harmony+MLP. Overall, we compare Cellcano to existing celltyping methods on their default and recommended settings. We do not think having an extra step of data integration added on to an existing celltyping method, for example, Harmony+EpiAnno, is necessary.

This being said, we have performed additional studies using different data integration methods with MLP for celltyping. Overall, these results are worse than Cellcano. Below we provide some details on these extra results. The new results have now been included and analyzed in the subsection "*Cellcano works better than prediction with batch effect removed*" under the Results section. Figures have also been updated (Figure 3A-C, Supplementary Figure S3-5) and added (Supplementary Figure S9-11). An additional subsection "*Integration with label transfer methods*" indicating how we perform the analysis has also been added to the Methods section.

We selected the top-performed integration methods from a benchmarking study (Luecken et al. 2022) where the authors evaluated 16 single-cell integration methods on scATAC-seq data. According to their benchmarking results, the top performers are LIGER (Welch et al. 2019) with peaks as input, ComBat (Johnson et al. 2007) with peaks as input, LIGER using genome-wide bins as input and ComBat using genome-wide bins as input. Besides LIGER and ComBat, we also included a recently published integration named Portal (Zhao et al. 2022), which is developed for atlas-level single-cell genomics integration. Based on the integration outputs, we performed MLP to do cell type prediction.

We summarized the 29 human PBMCs prediction experiments into a heatmap below where each row represents for one prediction experiment and each column represents for one method. We ordered the columns of the heatmap to make the leftmost column having the best performance gains. As shown in the figures, in general using integration method with MLP do not perform better than using the default setting of the celltyping method without data integration. Among all integration methods with MLP, Combat with either peaks or genome-wide bins as input rank the second. From the heatmap, Harmony+MLP performed the worst. We then performed the paired t-test to test the overall accuracy differences between Cellcano and Combat with peaks as input, the p-value is 0.0035. Same test has been performed to test the differences between Cellcano and Combat with genome-wide bins as input, the p-value is 0.0044. This indicates that Cellcano steadily outperforms all integration methods with label transferred.

Similarly, in 21 mouse brain experiments, ComBat with peaks as input ranks the second. We again performed a paired t-test between Cellcano and ComBat with peaks as input and the p-value is $6.144e-05$.

We have also shown data integration results in the low-dimensional space (Supplementary Figure S11) for a dataset. Even though the reference and target datasets or the individuals

seem to be well mixed after integration in the tSNE plots, the celltyping results are not necessarily better.

References:

- Ma, Wenjing, Kenong Su, and Hao Wu. "Evaluation of some aspects in supervised cell type identification for single-cell RNA-seq: classifier, feature selection, and reference construction." *Genome biology* 22.1 (2021): 1-23.
- Luecken, Malte D., et al. "Benchmarking atlas-level data integration in single-cell genomics." *Nature methods* 19.1 (2022): 41-50.
- Welch, Joshua D., et al. "Single-cell multi-omic integration compares and contrasts features of brain cell identity." *Cell* 177.7 (2019): 1873-1887.
- Johnson, W. Evan, Cheng Li, and Ariel Rabinovic. "Adjusting batch effects in microarray expression data using empirical Bayes methods." *Biostatistics* 8.1 (2007): 118-127.
- Zhao, Jia, et al. "Adversarial domain translation networks for integrating large-scale atlas-level single-cell datasets." *Nature Computational Science* 2.5 (2022): 317-330.

(Comment 7) Throughout the authors use very vague qualitative terms with no supporting statistics in the text. When saying a method performs "better" provide quantitative information.

We have added paired t-test results to show that Cellcano outperforms all other competing methods. These statistics have been added in the subsection "Cellcano outperforms existing supervised scATAC-seq celltyping methods" under the Results section.

(Comment 8) Claiming 50 experiments is misleading – the authors devised 50 means of assessing data from four PBMC and two mouse brain datasets.

Yes, your understanding is correct that all experiments are derived from six datasets including four human PBMCs datasets and two mouse brain datasets. Here, each experiment refers to a specific celltyping prediction task with different reference and target data. For example, we use different individuals in the human PBMC or mouse brain as reference and target. Thus, even with the handful datasets, we can design a lot of prediction experiments. Overall, we present four scenarios where celltyping can be applied:

1. Intra-dataset individual prediction: users have one confidently annotated scATAC-seq profile from one individual and want to use it to annotate all other individuals from the same study.
2. Inter-dataset individual prediction: users have one confidently annotated scATAC-seq profile from one individual and want to use it to annotate other individuals from different studies. In the mouse brain experiments, a special case is that we have experiment predicting cell types not only for a different subject but also for a different brain region because mouse brain has several brain regions. We count them into this category.
3. Inter-dataset prediction (combined reference): users have several well annotated scATAC-seq datasets and wish to use a large collection of public datasets to increase the reference data size and improve the prediction result. This is based on our previous research (Ma et al. 2021) where we found that combining individuals or datasets as reference could lead to better prediction results.
4. Inter-dataset prediction (combined target): users have scATAC-seq data from multiple batches and want to determine their cell types in one run using a given reference.

These scenarios are introduced in the subsection "Cellcano works better than prediction with batch effect removed" in the Results section. In our comparison, to fairly present that Cellcano

outperforms other methods, we also have another prediction scenario where we use FACS-sorted datasets as target. This can be regarded as validating prediction performances with ground-truth labels (inter-dataset prediction: ground truth).

We design experiments from the above categories and end up with 29 experiments from human PBMCs datasets and 21 experiments from mouse brain datasets in our paper. We provided details for all experiments in Supplementary Table S2. To make it clearer, we have added the designed experiments categories into the Supplementary Note S5.

References:

Ma, Wenjing, Kenong Su, and Hao Wu. "Evaluation of some aspects in supervised cell type identification for single-cell RNA-seq: classifier, feature selection, and reference construction." *Genome biology* 22.1 (2021): 1-23.

(Comment 9) The section on detailing why gene scores are used follows the overall performance analysis. This should precede the overall performance – same goes for the cellcano anchors section. The text in these could also be trimmed substantially.

I would suggest adding a paragraph each for the gene score choice and anchor properties within the cellcano framework section, summarizing the main points along with further detail in the methods section. Many details would make more sense there than in the main text.

Notably these sections have better reporting of quantitative metrics.

We have modified the subsection “*The Cellcano framework*” under the Results section to provide some discussion on the choice of inputs and the anchor cell properties. We also moved the two subsections “*The choice of using gene score as input*” and “*Properties of Cellcano anchors*” in front of the overall performance section. There are a lot of results in those two subsections that are important information for our method. Thus, we would like to keep the most important information in the Results section and put other parts in the Supplementary Note S2-4. We modified the language in those two subsections rather extensively to make them more concise.

(Comment 10) The “motivation for cellcano” section in the methods can be removed – this is just rehashing the intro and is not relevant to the methods.

Thank you for your suggestion. We have removed the section.

(Comment 11) The authors use tSNE, but often UMAP performs better on many datasets. It would be good to show both.

Thank you for your suggestions. During our explorations, we generated two sets of figures. From our observations, even though tSNE and UMAP have different appearances, they do not have significant differences in terms of conveying the idea. Below are some tSNE and UMAP visualizations from the human PBMC experiments and the mouse brain experiments. They show the dataset batch in the left panels and the cell labels in the right panels. The top two figures show tSNE visualizations and the bottom two figures show UMAP visualizations.

1. Use individual PBMC_D10T1 from Granja et al. human PBMCs dataset to predict cell types in another individual PBMC_Rep1 from Satpathy et al. human PBMCs dataset

2. Using 10X human PBMCs dataset to predict cell types in FACS-sorted human PBMCs dataset

3. Using all mice from Lauren et al. mouse brain dataset to predict cell types in PreFrontalCortex_62216 from Cusanovich et al. mouse brain dataset

4. Using mice individual WholeBrainA_62816 from Cusanovich et al. mouse brain dataset to predict cell types in mouse1 from Lauren et al. mouse brain dataset

After comparisons, we selected tSNE visualizations and used them as main figures in our manuscript. We now have added UMAP figures into Supplementary Figure S6A-C and Supplementary Figure S8D-F showing prediction performances from a human PBMCs experiment and a mouse brain experiment respectively.

(Comment 12) The Figure 2 panels D, E and F are hard to interpret. Cells are colored by ground truth, so how are they showing the prediction?

After the adjustment of the paper structure, the original Figure 2 has become the Figure 3 in the revised manuscript. Only cells in Figure 3D are colored by ground truth. In Figure 3E, cells are colored by first-round prediction results; in Figure 3F, cells are colored by the final prediction. We use Figure 3D as a ground-truth visualization comparison, and we show that some wrongly

assigned cells in Figure 3E are corrected after the second-round prediction in Figure 3F. The information is detailed in the figure legend for Figure 3 and the manuscript Results section “Cellcano outperforms existing supervised scATAC-seq celltyping methods”.

(Comment 13) In Fig 3 there is a lot of variability in model performance across the four datasets. Why?

This is a good but complicated question. Supervised prediction tasks, regardless of the task itself, on different reference and target data will have variabilities in the prediction results. These variabilities can be caused by many factors, for example, the signal-to-noise ratio (SNR) in the target data, differences in the reference and target data, etc. In our scATAC-seq celltyping problem, the differences can be caused by differences in cell-type-specific profiles, differences in cell type proportions, the training data size (number of cells in the reference data), number of cell types, etc. Overall, the variability can come from all above perspectives.

(Comment 14) There needs to be a figure diagramming what the experiments are for the comparisons. A schematic of datasets used, how they are processed, and the specific tests would help greatly.

Conceptually, the design for the experiment is rather straightforward. For each experiment, we just take one reference dataset with cell labels, and predict the cell labels in one target dataset. We have added a new subsection “Overall scheme of data processing and analysis” under the Methods section to make the whole process more coherent and clearer. This new section includes the content of the original subsection “Data Preprocessing by ArchR”, which provided details for data processing. We add the following figure to summarize the data preprocessing procedure. This schematic figure is now included as the Supplementary Figure S12. The details for all experiments are provided in the Supplementary Table S2, which lists reference and target datasets in all 50 experiments. Supplementary Note S6 provides details on how we processed each dataset.

(Comment 15) There are no plots showing the visualized brain data. Also the PBMC data shows very poor cluster separation...

We have added visualizations on one mouse brain prediction experiment where we use one individual from the Cusanovich et al. mouse brain dataset as reference to predict all cells from the Lareau et al. mouse brain dataset. This has been added as Supplementary Figure S8 and related analysis is added in the Results section “*Cellcano outperforms existing supervised scATAC-seq celltyping methods*” where we analyze the increased performance by using our second-round prediction.

The poor cluster separation of human PBMCs is caused by the nature of the data. In the original paper where the FACS-sorted human PBMCs dataset was published (Lareau et al. 2019), the Supplementary Figure 8(f) shows the original identified clusters for the FACS-sorted data, which is pasted below. Here, the figure (c) represents the clusters identified with biological knowledge from a collection of cells and (f) shows the projected FACS-sorted cells. In the original representation, the CD4 T cells, CD8 T cells and NK cells are clustered together without clear separation boundaries, while B cells and Monocytes are clearly separated. The cluster is done by *de novo* k-mer clustering methods from the ChromVAR paper. The same pattern is observed in our updated visualizations in Figure 3D.

Reference

Lareau, Caleb A., et al. "Droplet-based combinatorial indexing for massive-scale single-cell chromatin accessibility." *Nature Biotechnology* 37.8 (2019): 916-924.

(Comment 16) Figures need to be provided showing the harmony integration and how that method performs, as it is the most relevant. Essentially cellcano is just an integration method and not a celltyping method. It integrates then performs label transfer, but the integration is

where all of the relevance is.

As we responded before, Cellcano is not a data integration method. It does not try to remove the distribution discrepancies between reference and target datasets, and it does not transform either the reference or target data. Instead, our main purpose is celltyping and we do it without integrating the data. We showed Harmony integration performance in original Supplementary Figure S10-S11, where we use summarized gene activity scores as input. Now we have added the batch effect removal visualizations for all integration with label transfer methods into the newly added Supplementary Figure S11. Even though certain methods can remove batch effect, the cell type prediction is not necessarily better compared to Cellcano. Related discussions have been added in the subsection “*Cellcano works better than prediction with batch effect removed*” under the Results section.

Response to Reviewer 2’s comments:

In this paper, the authors present a supervised approach to catalog cells when only scATAC-seq data is available. The model uses existing pipelines (Archr) to compute gene level statistics (gene scores). Then, it selects features through an statistical test and train a MLP. If the target set is big enough, cellcano performs a second round of annotations in which it selects anchors (based on an entropy determination of the confidence of the labels) and trains a knowledge-distiller model. To assess the confidence and accuracy of the proposed approach, the authors benchmark cellcano against 6 competing approaches, used 7 metrics and plenty of data sets.

One of the weak points of the approach is the calculation of gene scores for cell annotation. The authors acknowledge this point, compare against different methods that summarize gene information and demonstrate that theirs is a good choice. Additionally, for the second round of annotations, cellcano uses anchors selected on confidence of annotation of the training dataset.

The method is available online and detail documentation on how to install it and how to run is also available.

More generally, the approach is simple and uses a common algorithm, MLP, to perform the annotation. Given the results, the method is sound and seems to be a top-performer. However, there are many approaches performing the same task already so it lacks novelty, this is also evident by the many approaches they compared against. Using a KD model to address the distributional-shift is a clever and novel approach to address one of the fundamental problems in single-cell genomics, batch and condition correction. Due to this fact, I believe the method merits publication. In addition, the paper is well written and does not need major revisions.

We greatly appreciate the positive comments. However, we would argue that there are not “*many approaches performing the same task already*” for scATAC-seq data. As of now, only EpiAnno is developed for supervised celltyping in scATAC-seq using scATAC-seq data as reference. Besides EpiAnno, only two other methods (Seurat and scJoint) use scRNA-seq data as reference and transfer cell labels to scATAC-seq datasets. To make the comparisons more comprehensive, we even included some scRNA-seq methods such as SingleR and ACTINN in the comparison.

Having said this, it will be great if the authors describe in more detail their intuition (hopefully supplemented by any mathematical proof) about the following topics (it can be done in the response letter).

(Comment 1) Why does Cellcano and sc-joint perform similarly when they are based on completely different approaches?

Thanks for the question. Cellcano and scJoint performs similarly when using the FACS-sorted data as target (Figure 3A in the updated manuscript), but not that similar in human PBMCs and mouse brain datasets (Figure 3B-C in the updated manuscript). Overall, scJoint is the second-best performer among all methods. We think it is common when two different methods have similar results. For example, SVM and random forest would perform similarly in many tasks, even though they have intrinsically different algorithms.

Moreover, the boxplots provide the marginal distributions of all accuracies, which represents the overall performances. We have performed additional analyses for a pairwise comparison on both methods' prediction performances in each experiment and checked whether they show similar trends. Below are the line plots for three metrics in the 29 human PBMCs experiments where the solid line represents Cellcano's performance, and the dotted line represents scJoint's performance. We then perform a paired t-test and the p-values of Accuracy, ARI, and macroF1 between Cellcano and scJoint are 0.008, 0.050 and 0.040 respectively, indicating that Cellcano performs better.

Similarly, we plotted the lines for three metrics in the 21 mouse brain experiments. The paired t-test p-values of Accuracy, ARI, and macroF1 between Cellcano and scJoint are 0.012, 0.005 and 0.042 respectively.

Although for some experiments Cellcano and scJoint have close prediction performances, overall, there are still significant differences between the two methods, where Cellcano has

better performances. We have also added all original prediction performances into heatmaps (Supplementary Figure S9-10) categorized by different experimental categories. From the heatmaps, it is much easier to see the performance variations.

(Comment 2) Why does cellcano needs 2 rounds of predictions? Why is an MLP not enough to perform annotations in a first pass? If the intuition is that cells are incorrectly labeled at the decision boundary, can you plot the label uncertainty next to the annotations before and after KD?

The reason why we design the two-round prediction strategy is because of the distribution discrepancy between the reference and target data. Such discrepancy could cause inaccurate prediction in an MLP. Our second-round trains a classifier from the target data so that the classifier can best capture target data distribution and provide better prediction results. In our comparisons, ACTINN can represent the performance of the first-round prediction using MLP, since it is based on a vanilla MLP. As shown in Figure 3A-C (in the updated manuscript), Cellcano can achieve better prediction performances than ACTINN.

Yes, you are correct that there can be cells wrongly labeled at the decision boundary. During the exploration of the entropy, we noticed that the tSNE coordinates had some randomness and thus resulted in tSNE visualizations slightly different from our first submission. But we assure that the figures still convey the same message and what have been seen in the original figures can still be seen in the newly generated tSNE figures. The randomness is caused by the scanpy package and there are many Github issues discussing about it. We now stored all the tSNE coordinates for all experiments for future reproducibility.

Here, we attached the newly generated Figure 3D-F (in the updated manuscript) to first show cell type information in the target dataset (FACS-sorted human PBMCs dataset).

We then showed the comparison of prediction entropy of cells before and after KD. As expected, the cells on the boundary are unconfidently predicted before the KD model, while after the KD, most of the cells are more confidently predicted with lower entropies.

We also plotted the predicted probabilities for CD8 T cells before and after KD. Before KD, CD8 T cells are predicted with lower probabilities compared to those after KD, especially for the area where CD8 T cells are corrected during our second-round prediction.

These above figures have now been added to the Supplementary Figures as Figure S7 and discussions have been added to the subsection “*Cellcano outperforms existing supervised scATAC-seq celltyping methods*” under the Results section.

(Comment 3) *Can you describe if there could exist incorrect anchors with high confidence that would produce incorrect results? Can this happen if the number of cells in target and reference data set have completely different cell numbers?*

This is a very good question. Yes, we do observe that there can be incorrect anchors with high confidence. We plot the anchor accuracy in the confusion matrices below. Here, we used the same experiment shown in Figure 3D (one FACS-sorted human PBMCs dataset as target, a combination of four individuals as reference). As shown in the confusion matrix, there is around 20% of the NK cells wrongly assigned to CD8 T cells. This is reasonable because there are no clear separation boundaries between CD8 T cells and NK cells. The metrics for the anchors are 0.941, 0.877 and 0.935 for Accuracy, ARI and macroF1 respectively.

However, it is unclear to us what the reviewer is asking in the second sentence “*Can this happen if the number of cells in target and reference data set have completely different cell numbers?*” We therefore interpreted it in two ways:

1. The relationship between anchor accuracy and data sizes of reference and target datasets.
2. The relationship between anchor accuracy and cell type proportion differences between reference and target datasets.

Under our first interpretation, we have conducted analysis between anchor accuracy and the data sizes (number of cells) of reference and target datasets for both human PBMCs and mouse brain experiments. We first correlate the logit transformed anchor accuracy with reference data size, target data size and fold change between reference and target datasets. In 29 human PBMCs experiments, we noticed that there existed a negative correlation (Pearson Correlation Coefficients: -0.443, p-value: 0.016) between logit(accuracy) and the fold change between reference and target data sizes. While for the reference size and target size, the Pearson Correlation Coefficients are 0.168 (p-value: 0.384) and -0.232 (p-value: 0.226), all non-significant.

We then fit a linear model by robust regression with logit(accuracy) as the response variable and the reference data size, target data size and the fold change between reference and target data sizes as explanatory variables. The estimated coefficients of reference data size and target data size are all 0s and the estimated coefficient of the fold change is -0.122 (p-value: 0.013).

We performed the same analysis in 21 mouse brain experiments but did not observe any significant correlations. The Pearson Correlation Coefficients are -0.283 (p-value: 0.214), 0.395 (p-value: 0.076) and 0.286 (p-value: 0.208) between logit(accuracy) and reference data size, target data size and the fold change between reference and target sizes respectively. The

robust linear regression has no significant result, even though the estimated coefficient of the fold change is still negative: -0.137 (p-value: 0.252).

We also combine the 29 human PBMCs experiments and 21 mouse brains experiments together and fit the robust linear regression model with $\text{logit}(\text{accuracy})$ as the response variable. Since we combine two different systems with each having different baseline performances, we add a categorical explanatory variable to represent the systems. The robust linear regression result shows that the estimated coefficient of the fold change is -0.153 (p-value: 0.039). Overall, these results suggest that the fold changes of cell numbers in reference and target data could have mildly negative impact on the anchor accuracies.

Under our second interpretation, we analyzed whether anchor accuracy can be different when the cell type proportions of reference and target datasets are the same. We used the same experiments and performed the same correlation and robust regression analyses on using $\text{logit}(\text{accuracy})$ as response variable and cell type proportion differences as explanatory variable. We summarized the cell type proportions differences between the reference and target datasets with the mean square calculation.

In both human PBMCs (left panel) and mouse brains (right panel) systems, we found negative correlation with Pearson Correlation Coefficients as -0.122 (p-value: 0.527) and -0.489 (p-value: 0.025) respectively. We also performed the robust regression analyses where the results show the estimated coefficients are -8.555 (p-value: 0.369) and -467.750 (p-value: 0.021) for human PBMCs and mouse brains experiments.

We also combined all experiments together and added biological system indicator as a categorical explanatory variable into the robust linear regression model. The results show that the cell type proportion differences have an insignificant impact on the anchor accuracies (p-value: 0.504).

Moreover, we empirically tested whether cell type proportion can affect anchor accuracies. We therefore take one experiment (one FACS-sorted human PBMCs dataset as target, a combination of four individuals from Satpathy et al. PBMCs dataset as reference) as example. We resample the reference dataset to let them have the same cell type proportion as the target

dataset while remaining the original reference data size. With the adjusted reference dataset, we performed Cellcano. The anchors prediction performance is shown below. The metrics of anchors are 0.942, 0.876 and 0.938 for Accuracy, ARI and macroF1 respectively, where the anchors' performances are similar to previous performances which are 0.941, 0.877 and 0.935. At least in this case, turning the reference and target datasets to have the same cell type proportions does not help with anchor accuracy.

In the reference dataset, the proportion of cell types are:

Monocytes	CD4 T cells	CD8 T cells	B cells	NK cells
0.292	0.287	0.233	0.101	0.086

While in the target dataset, the proportion of cell types are:

Monocytes	CD4 T cells	CD8 T cells	B cells	NK cells
0.127	0.240	0.223	0.216	0.194

Overall, these results show that the “number of cells in target and reference data” does not have significant impact on the anchor accuracy or the final prediction results.

(Comment 4) Minor: This sentence does not make sense. Variational inference is not used as a purpose, it is a method to perform approximated posterior inference.

"ScANVI is a semi-supervised learning method which uses deep generative model for variational inference purpose to first integrate scRNA-seq datasets and then transfer annotations"

Thank you for pointing it out. During the revision, we noticed that scANVI had a prior assumption on the input data and it would be inappropriate to use in scATAC-seq celltyping. We therefore removed our comparison to scANVI.

REVIEWER COMMENTS

Reviewer #1 (Remarks to the Author):

The authors have addressed many of my comments; my major component was that I saw it as an integration method which the authors have clarified to a degree; however much of it is semantic. While Cellcano does not seek to integrate and transform a dataset; it deploys many of the same techniques as integration methods where cells from one dataset are anchored to another which one could argue is the core of integration. Rarely the data is transformed after this and the cross-modality cell anchors are what is leveraged (ie cell type matching) and the original data for those cell types are used. From that stance I see little difference between a celltyping method that anchors between two datasets for cell type assignment vs one that does the same and calls itself integration. (ie we use 'integration' between an annotated ATAC dataset and unannotated ATAC dataset frequently, where we leverage the anchors from signac or ArchR and label transfer to the unannotated dataset)

On that note - there is still no comparison to taking the approach I mentioned - i.e. 'integration' between an annotated and unannotated ATAC dataset using Signac as well as ArchR, both being tools anyone analyzing scATAC data will be using, and then label transfer of cell types. This is the default to what people are doing now and just because those are 'integration' the purpose of application and results are the same.

The terminology of experiments will be confusing for a majority of reviewers who will think it is referring to individual datasets / bench experiments. I strongly suggest changing this terminology to be more suitable to the broad readership of the journal.

As a user of these methods, there is nothing that makes this method jump out to me, which I believe is more of a presentation issue. Is there any way to simply show a UMAP/tSNE of an unannotated dataset and one of the reference used, then show the cell type transfer labeling plus the actual annotations? This would clarify things for potential users and make it more approachable and appealing to them versus comparisons with a lot of methods they probably are not using.

Reviewer #2 (Remarks to the Author):

The authors have addressed my concerns.

Introduction

We thank the reviewers for bringing up further concerns and providing the thoughtful comments for us to improve our manuscript.

Based on the reviewers' comments, we have carefully revised the manuscript (modifications in the manuscript are highlighted in **red font**). In this revision, we made following major modifications:

1. We have added Signac's performances into the benchmarking comparisons.
2. We have changed the "experiments" term to "celltyping tasks".
3. We have provided visualization results on celltyping with reference datasets information.

We addressed the individual points raised by the reviewers, as detailed below. The reviewers' comments are in *italic blue*, and our replies are in regular font.

Response to Reviewer 1's comments:

The authors have addressed many of my comments; my major component was that I saw it as an integration method which the authors have clarified to a degree; however much of it is semantic. While Cellcano does not seek to integrate and transform a dataset; it deploys many of the same techniques as integration methods where cells from one dataset are anchored to another which one could argue is the core of integration. Rarely the data is transformed after this and the cross-modality cell anchors are what is leveraged (ie cell type matching) and the original data for those cell types are used. From that stance I see little difference between a celltyping method that anchors between two datasets for cell type assignment vs one that does the same and calls itself integration. (ie we use 'integration' between an annotated ATAC dataset and unannotated ATAC dataset frequently, where we leverage the anchors from signac or ArchR and label transfer to the unannotated dataset)

On that note - there is still no comparison to taking the approach I mentioned - i.e. 'integration' between an annotated and unannotated ATAC dataset using Signac as well as ArchR, both being tools anyone analyzing scATAC data will be using, and then label transfer of cell types. This is the default to what people are doing now and just because those are 'integration' the purpose of application and results are the same.

We are glad that we have addressed most of the comments. For the major concern, we perhaps have different comprehensions on the definition of "data integration". We understand the reviewer's point that cell type identification is some kind of data integration, since the cell type label information are transferred between reference and target datasets. However, we think the term "data integration" mostly represents a family of approaches to remove distributional discrepancy among datasets from different studies, platforms, or modalities, and then combine the information together to perform certain tasks. The goals of the data integration are not limited to cell label transfer, they can also be missing data imputation, modality alignment, or joint embedding learning. According to a recent atlas-level integration benchmark study (Luecken et al. 2022), "data integration" methods are developed to "combine high-throughput sequencing datasets or samples to produce a self-consistent version of the data for downstream analysis". Similar to that, another benchmark study on batch-effect correction (Tran et al. 2021) use the terms "batch-effect removal" and "data integration" interchangeably. Unlike the reviewer

mentioned that “rarely the data is transformed”, many data integration methods do some data transformation, either in the original or latent space.

Different from “data integration”, “celltyping” purely aims to train a classifier for cell type identification and does not necessarily transform data. When distributional discrepancies are strong among datasets (i.e., in scATAC-seq datasets), traditional supervised classifiers can be greatly affected. Therefore, some methods perform “data integration” before label transfer, but the “integration” is just an intermediate step. In our method, the celltyping is performed without data integration, or more precisely, without *data transformation*.

We acknowledge that there are some scATAC-seq pipelines providing both “integration” and “celltyping” functions, such as Signac. However, their main purpose is to provide end-to-end functionalities such as quality control, data preprocessing, dimension reduction and advanced downstream analysis. We have added the “celltyping” results from Signac, following their Vignettes (https://stuartlab.org/signac/articles/integrate_atac.html). In human PBMCs celltyping tasks, when using FACS-sorted data as target, Cellcano greatly outperforms Signac. When using silver-standard data as target, Cellcano ranks first in Accuracy and ARI while ranks the second in macroF1 where Signac ranks first. When performing a paired t-test between Cellcano and Signac on all tasks, Cellcano significantly outperforms Signac in Acc (p-value: 0.039) and ARI (p-value: 0.012).

In mouse brain celltyping tasks, we again notice that Cellcano outperforms Signac.

Taken altogether, Cellcano outperforms all other methods. We now have added Signac into comparison and updated corresponding figures in Figure 3A-C, Figure 4, Figure S3-5, and Figure S9-10.

Note that there are several difficulties using Signac to perform “celltyping”. First, Signac only accepts scATAC-seq fragment files. When the input data is bam files, users must use another tool [sinto](https://timoast.github.io/sinto/installation.html) (<https://timoast.github.io/sinto/installation.html>) to transform bam files into fragment files. Second, for each celltyping prediction, users need to first call peaks on the reference dataset and then generate peak counts for both reference and target datasets. This usually takes a long time to accomplish (~1-2h for processing reference and target peak counts). Third, since Signac needs to project target dataset onto reference dataset to compute integration anchors, users need to run Signac every time for a new celltyping prediction. Compared to Signac, Cellcano uses gene scores as input, which (1) achieves better celltyping accuracy and computational efficiency; (2) can take pre-trained classifier on atlas-level reference dataset as input; and (3) can be easily connected to scRNA-seq and other modalities.

References:

Luecken, Malte D., et al. "Benchmarking atlas-level data integration in single-cell genomics." *Nature methods* 19.1 (2022): 41-50.

Tran, Hoa Thi Nhu, et al. "A benchmark of batch-effect correction methods for single-cell RNA sequencing data." *Genome biology* 21.1 (2020): 1-32.

The terminology of experiments will be confusing for a majority of reviewers who will think it is referring to individual datasets / bench experiments. I strongly suggest changing this terminology to be more suitable to the broad readership of the journal.

Thank you for your suggestions. We have changed all the “experiments” to “celltyping tasks” to avoid potential confusions.

As a user of these methods, there is nothing that makes this method jump out to me, which I believe is more of a presentation issue. Is there any way to simply show a UMAP/tSNE of an unannotated dataset and one of the reference used, then show the cell type transfer labeling

plus the actual annotations? This would clarify things for potential users and make it more approachable and appealing to them versus comparisons with a lot of methods they probably are not using.

Thank you for your suggestion. We have shown the predicted cell types of target dataset in Figure 3D-F where 3D is the ground-truth label and 3F is the final prediction. Here, according to your suggestions, we added the reference dataset into the tSNE plots and provided comparisons on how other methods perform. As shown in the following figures, the existence of batch effect between reference and target datasets indicates that we do not perform any data transformation.

We first plotted the batch information (left panel) and their ground-truth cell types (right panel). As shown in the left panel, there exists a strong domain shift between reference and target datasets.

Then, we plotted all predicted cell types from different methods. The prediction accuracies of the following methods are:

Cellcano	Signac	Seurat	scJoint	SingleR	ACTINN	EpiAnno
0.900	0.829	0.686	0.903	0.829	0.869	0.835

As for integration methods with label transfer, we have provided the integrated embedding space along with cell type information in Supplementary Figure S11.

In summary, we believe our supervised celltyping method is accurate, efficient, and scalable and we hope it can replace the traditional “clustering + annotation with prior knowledge” step in all single-cell genomics data analysis. We believe the correctly predicted cell types can better guide the scATAC-seq data preprocessing steps such as identifying cell-type-specific accessibilities, peak-to-gene linkage, etc.

Besides the accuracy, we aim to have a pre-trained classifier for publicly available high-quality datasets which can be quickly adapted to predict cell types in the target dataset.

Response to Reviewer 2’s comments:

The authors have addressed my concerns.

Thank you for your previous comments and we are glad that we have addressed them.

REVIEWERS' COMMENTS

Reviewer #1 (Remarks to the Author):

I appreciate the additional comparisons as well as clarifications by the authors, all of my comments have been fully addressed. I believe we are now 'in sync' with the presentation and characterization of cellcano and I have no further comments.

Introduction

We thank the reviewers for their thoughtful comments and suggestions for us to improve our manuscript. We addressed the individual points raised by the reviewers, as detailed below. The reviewers' comments are in *italic blue*, and our replies are in regular font.

Response to Reviewer 1's comments:

I appreciate the additional comparisons as well as clarifications by the authors, all of my comments have been fully addressed. I believe we are now 'in sync' with the presentation and characterization of cellcano and I have no further comments.

Thank you for your previous comments and we are glad that we have addressed them.